# Boulton-Katritzky Rearrangement of 5-Substituted Phenyl-3-[2-(morpholin-1-yl)ethyl]-1,2,4-oxadiazoles as a Synthetic Path to Spiropyrazoline Benzoates and Chloride with Antitubercular Properties

**DOI:** 10.3390/molecules26040967

**Published:** 2021-02-12

**Authors:** Lyudmila Kayukova, Anna Vologzhanina, Kaldybai Praliyev, Gulnur Dyusembaeva, Gulnur Baitursynova, Asem Uzakova, Venera Bismilda, Lyailya Chingissova, Kydyrmolla Akatan

**Affiliations:** 1JSC A. B. Bekturov Institute of Chemical Sciences, 106 Shokan Ualikhanov St., Almaty 050010, Kazakhstan; praliyevkd@mail.ru (K.P.); g_gazinovna@mail.ru (G.D.); guni-27@mail.ru (G.B.); a7_uzakova@mail.ru (A.U.); 2A. N. Nesmeyanov Institute of Organoelement Compounds, Russian Academy of Sciences, 28 Vavilov St., В-334, 119991 Moscow, Russia; 3National Scientific Center of Phthisiopulmonology of Ministry of Health of the Republic of Kazakhstan, 5 Bekkhozhin St., Almaty 050010, Kazakhstan; venerabismilda@mail.ru (V.B.); lchingisova@mail.ru (L.C.); 4National scientific laboratory, S. Amanzholov East Kazakhstan State University, 18/1 Amurskaya St., Ust-Kamenogorsk 070002, Kazakhstan; ahnur.hj@mail.ru

**Keywords:** 1,2,4-oxadiazoles, spiropyrazolinium compounds, in vitro antitubercular screening, X-ray diffraction, molecular docking

## Abstract

The analysis of stability of biologically active compounds requires an accurate determination of their structure. We have found that 5-aryl-3-(2-aminoethyl)-1,2,4-oxadiazoles are generally unstable in the presence of acids and bases and are rearranged into the salts of spiropyrazolinium compounds. Hence, there is a significant probability that it is the rearranged products that should be attributed to biological activity and not the primarily screened 5-aryl-3-(2-aminoethyl)-1,2,4-oxadiazoles. A series of the 2-amino-8-oxa-1,5-diazaspiro[4.5]dec-1-en-5-ium (spiropyrazoline) benzoates and chloride was synthesized by Boulton–Katritzky rearrangement of 5-substituted phenyl-3-[2-(morpholin-1-yl)ethyl]-1,2,4-oxadiazoles and characterized using FT-IR and NMR spectroscopy and X-ray diffraction. Spiropyrazolylammonium chloride demonstrates in vitro antitubercular activity on DS (drug-sensitive) and MDR (multidrug-resistant) of MTB (*M. tuberculosis*) strains (1 and 2 µg/mL, accordingly) equal to the activity of the basic antitubercular drug rifampicin; spiropyrazoline benzoates exhibit an average antitubercular activity of 10–100 μg/mL on MTB strains. Molecular docking studies revealed a series of *M. tuberculosis* receptors with the energies of ligand–receptor complexes (−35.8–−42.8 kcal/mol) close to the value of intermolecular pairwise interactions of the same cation in the crystal of spiropyrazolylammonium chloride (−35.3 kcal/mol). However, only in complex with transcriptional repressor EthR2, both stereoisomers of the cation realize similar intermolecular interactions.

## 1. Introduction

There are pyrazoline-containing compounds that act as active pharmaceutical ingredients of such commercially available drugs as *aminopyrine* (*aminophenazone*; analgesic and antipyretic), *dipyrone* (*metamizole*, *noramidopyrine*; analgesic), *antipyrine* (*benzocaine*; non-narcotic analgesic, an antipyretic and antirheumatic), *zaleplon*(hypnotic and sedative), *celecoxib* (*Aclarex*, *Celebrex*; anti-inflammatory and antirheumatic drug), *allopurinol* (uricostatic agent, xanthine oxidase inhibitor) [1]. Therefore, there is always a demand for new molecules, methodologies and improved synthetic approaches to novel pyrazoline derivatives.

Pyrazolines, as noticeable, practically meaningful nitrogen-containing heterocyclic compounds, can be synthesized by a variety of methods. However, one of the most popular methods is the Fischer and Knoevenagel synthesis based on the reaction of *α,β*-unsaturated ketones with phenylhydrazine in acetic acid under refluxing conditions. However, depending on the reactivity of molecules and the need of the chemist, they had synthesized the pyrazolines under different solvent media and acidic or basic conditions [2,3,4].

Information on pyrazolinium structures with a quaternary nitrogen atom is limited. Thus, two examples of biologically active pyrazolinium salts were found: 3-amino-1-ethyl-1-phenyl-4,5-dihydro-1*H*-pyrazolinium iodide (PubChemCID: 13585073 structure, [5]) and 3-amino-1,4-dimethyl-1-phenyl-2-pyrazolinium iodide (PubChemCID: 13064197 structure, [6]) (Scheme 1):

In some works, we found spiropyrazolinium compounds with a quaternary nitrogen atom, which is common for two heterocycles. When studying the stability of 3-(2-aminoethyl)-5-aryl-1,2,4-oxadiazoles having six-membered cyclic tertiary 2-amino groups towards hydrolysis, we found that they were capable of rearranging to spiropyrazoline benzoates or chlorides [7,8,9].

Particularly, upon keeping 3-[2-(4-phenylpiperazin-1-yl)ethyl]-5-phenyl-1,2,4-oxadiazole recrystallized in 2-PrOH under conditions of air moisture access for nine months for growing single crystals for X-ray structural analysis or by exposure of 3-[2-thiomorpholin-1-yl)ethyl]-5-aryl-1,2,4-oxadiazoles in ethanol with ethereal HCl solution the above-mentioned 1,2,4-oxadiazoles underwent the rearrangement to spiropyrazolinium benzoates or chlorides (Scheme 2) [7,8]: 

Furthermore, at targeted exposure on 3-[2-(piperidin-1-yl)ethyl]-5-aryl-1,2,4-oxadiazoles with: (*i*) water, (*ii)* water in DMF or (*iii*) ethereal HCl they underwent rearrangement with the formation 2-amino-1,5-diazaspiro[4.5]dec-1-en-5-ium benzoates or chlorides (Scheme 2). In the latter case, along with the formation of spiropyrazolinium chloride hydrate from 3-[2-(piperidin-1-yl)ethyl]-5-(3-chloro-phenyl)-1,2,4-oxadiazole hydrochlorides of starting 1,2,4-oxadiazoles were obtained as secondary products [9].

These facts are consistent with the known for 3,5-substituted 1,2,4-oxadiazoles with a saturated side chain spontaneous thermally induced monomolecular Boulton–Katritzky rearrangement and provided the first examples of spirocompound formation through such reaction. In general, a variety of Boulton–Katritzky rearrangements could be represented as the following scheme (Scheme 3) [10,11]:

In addition, spiropyrazolinium structures–2-amino-8-oxa-1,5-diazaspiro[4.5]dec-1-ene-5-ammonium arylsulfonates are formed at arylsulfochlorination of β-aminopropioamidoximes (Scheme 4) [12].

The practical interest in the class of β-aminopropioamidoxime derivatives is supported by their pronounced local anesthetic, antitubercular, and antidiabetic activities [13,14,15,16].

Herein we report on the stability of 5-aryl-3-[β-(morpholin-1-yl)ethyl]-1,2,4-oxadiazoles towards hydrolysis at: (*i*) DMF with the two equivalent amount of water when heated to 60–70 °C; (*ii*) alcohol/ethereal HCl mixture. A number of previously unknown spiropyrazolinium salts were obtained and characterized using FT-IR and NMR spectroscopy and X-ray diffraction. In vitro antitubercular screening of spiropyrazoline benzoates and chloride was carried out, and their molecular docking was performed. It was shown that the hydrolysis of 1,2,4-oxadiazoles with a 3-morpholinoethyl substituent leads to spiropyrazoline compounds within 25–40 h. Acid hydrolysis of 1,2,4-oxadiazoles occurs immediately after reagents adding. In vitro antitubercular screening of benzoates and chloride of spiropyrazoline drug-susceptible and multidrug-resistant strains, *M. tuberculosis* revealed compounds with significant activity, and the results are in accordance with molecular docking studies.

## 2. Results and Discussion

### 2.1. Synthesis and Spectra

The synthesis of the starting compounds **1**, **2a**–**e**, **3a**–**e** and **4a**–**e** was described earlier [17]. 5-Substituted phenyl-3-[2-(morpholin-1-yl)ethyl]-1,2,4-oxadiazoles (**4a**–**e**) were obtained by heating of О-aroyl-(β-morpholin-1-yl)propioamidoximes (**3a**–**e**) in DMF at 70 °C for several hours, evaporating off the solvent in an oil pump vacuum and treating of the residue with acetone. 1,2,4-Oxadiazoles (**4a**–**e**) are obtained as crystalline precipitates from acetone (Scheme 5).

Physicochemical, FT-IR and NMR spectral characteristics of representatives of 1,2,4-oxadiazoles of the morpholine series (**4a**–**e**) were recorded immediately after isolation from DMF, and they correspond to the structure of 1,2,4-oxadiazoles. However, the X-ray diffraction analysis of single crystals grown for nine months from 1,2,4-oxadiazoles **4c**–**е** recrystallized from 2-PrOH showed a complete transition of 1,2,4-oxadiazoles into rearranged spiropyrazolinium compounds **5c**–**е** (Section 2.3.). This indicates the hydrolysis of 1,2,4-oxadiazoles **4а**–**е** by way of Boulton–Katritsky rearrangement to spiropyrazolinium compounds **5а**–**е** under the influence of air moisture. The rearrangement products **5a**–**e** have increased values of the mobility index *R*_f_ and m.p. in comparison with the initial compounds **4a**–**e** (Table 1).

Further in this work, we investigated the conditions for the Boulton–Katritsky rearrangement of 5-substituted phenyl-3-[2-(morpholin-1-yl)ethyl]-1,2,4-oxadiazoles (**4a**–**e**) under the deliberate establishment of hydrolysis conditions: (*i*) DMF with the two equivalent amount of water when heated to 60–70 °C; (*ii*) alcohol/ethereal HCl mixture in the presence of air moisture (Scheme 5).

As can be seen from Table 1, the heating time has an increased value (40 h) for electron-donor substituents in the phenyl ring of 1,2,4-oxadiazoles **4а**, **4b** in comparison with 1,2,4-oxadiazoles with an unsubstituted phenyl ring and with a phenyl ring having electron-withdrawing substituents—**4c**–**e** (25 h). Spiropyrazolinium compounds **5а**–**е** were obtained after evaporation of DMF in an oil pump vacuum, treatment of the residue with acetone with the isolation of rearranged products **5а**–**е** and their recrystallization from 2-PrOH. In the case of the action of ethereal HCl solution on alcohol solutions of 1,2,4-oxadiazoles **4a**–**e** in all cases, 2-amino-8-oxa-1,5-diazaspiro[4.5]dec-1-en-5-ium chloride (**6**) and the corresponding benzoic acids were isolated.

IR spectra view of spirocompounds **5a**–**e** and **6** differ from the IR spectra of 1,2,4-oxadiazoles **4a**–**e**. First, the former compounds have symmetric and asymmetric ν(N–H) stretching bands at 3152–3485 and 3158–3457 cm^−1^, respectively; second, there are pronounced bands of asymmetric and symmetric stretching vibrations of strong intensity ν(СОО*^–^*) at 1545–1557 cm^−1^ and 1420–1442 cm^−1^, respectively for **5a**–**e** and no stretching bonds of aromatic protons for salt **6**.

The ^1^H-NMR spectra of 1,2,4-oxadiazole **4a**–**e** were recorded immediately with isolation; if they were recorded after 1–2 weeks, then the emergence and increase in the intensity of the NH_2_ group signal of the rearranged spiropyrazolinium products **5a**–**e** in the region of δ 7.51–7.7.57 ppm was observed. It indicates a transition of 1,2,4-oxadiazole to the spiropyrazolinium compounds **4a**–**e**→**5a**–**e** in the presence of air moisture.

Comparison of the NMR spectra (^1^H and ^13^C) of compounds **4a**–**e** and **5a**–**e** shows almost no differences in the regional characteristics of the groups of protons and carbon atoms of the structures of 5-substituted phenyl-3-[2-(morpholin-1-yl)ethyl]-1,2,4-oxadiazoles (**4a**–**e**) and of 2-amino-8-oxa-1,5-diazaspiro[4.5]dec-1-en-5-ium benzoates (**5a**–**e**). Hence, protons of α-СH_2_ and β-СH_2_ groups in the first case give signals in the range δ 3.15–3.17 ppm and δ 3.65–3.66 ppm and in the region δ 3.15–3.19 ppm and δ 3.64–3.66 ppm—in the second. The signals of the aromatic protons of *para*-substituted 1,2,4-oxadiazoles **4a**, **4****b**, **4****d** and spiropyrazolinium benzoates **5a**, **5****b**, **5****d** have the form of two symmetric doublets with the spin–spin coupling constant *J* equal 7.5 and 8.0 Hz. Aromatic protons of 1,2,4-oxadiazoles and spiropyrazolinium benzoates with unsubstituted and *meta*-substituted phenyl rings have signals at δ 7.23–7.83 ppm (**4c**) and 7.22–7.80 ppm (**4e**) and 7.25–7.78 ppm (**5c**, **5****e**). Proton-containing substituents CH_3_O and CH_3_ have singlet signals with an intensity of 3 protons at δ 3.73 and 2.27 ppm for compounds **4a**, **4****b** and **5a**, **5****b**.

A distinctive feature of ^1^H-NMR spectra of benzoates **5a**–**e** from the spectra of 1,2,4-oxadiazoles **4a**–**e** is the presence of NH_2_ proton signal with the integral intensity of 2H at δ 7.51–7.57 ppm.

The ^1^H-NMR spectrum of spirocompound **6** contained the triplet proton signals of α- and β-CH_2_ groups at δ 3.16 and 3.68 ppm and signals of N(+)(CH_2_)_2_(CH_2_)_2_О and N(+)(CH_2_)_2_(CH_2_)_2_О groups at δ 3.40 and 3.92 ppm, respectively; no aromatic proton signals were observed.

Of the remarkable features of the ^1^H-NMR spectra of compounds **4a**–**e**, **5a**–**e**, and **6**, is that the axial and equatorial protons of the methylene groups located at the nitrogen atom of the morpholine ring give independent multiplet signals. In one case, these signals are superimposed with the common signal of two groups methylene protons located at the oxygen atom of the morpholine ring at δ 3.91–3.93 ppm with a total intensity of six protons and in the other case have a multiplet signal at ~δ 3.40 ppm intensity of two protons. The diastereotopicity of discussed geminal protons of compounds **4a**–**e**, **5a**–**e**, and **6** is associated with a dynamic cause due to slow rotation of the morpholine heterocycle. The effect of hindered inversion of six-membered heterocycles, with a chair-like conformer with fixed positions of the axial and equatorial protons being predominant, in the ^1^H-NMR spectra is a known fact reported in reference data [18]. In addition, the diastereotopicity of these geminal protons of compounds **5a**–**e** and **6** is associated with asymmetry due to the presence of the spirocyclic system.

In the ^13^C spectra of the compounds **4a**–**e**, **5a**–**e** and **6,** all signals of aliphatic and aromatic protons were recorded in the expected regions. 

So, in the ^13^C NMR spectra the characteristic groups of compounds **4a**–**e**, **5a**–**e** and **6** include the signals of: α-methylene groups at δ 31.4 ppm; β-methylene groups at δ 62.1 ppm; signals of carbon atoms of methylene groups with intensity 2C located at nitrogen and oxygen atoms of the heterocycle are in the regions δ 62.4–62.5 ppm and δ 63.2–63.3 ppm. Two signals of carbon atoms of C=N bonds of 1,2,4-oxadiazoles **4****а**–**е** are in the regions δ 167.0–168.9 ppm and δ 169.2–169.5 ppm. The carbon atoms of the C=N bond of the pyrazoline ring of compounds **5****а**–**е** have two signals at δ 167.0–168.8 ppm and δ 169.2 ppm, which may be due to the existence of enantiomers A and B. The carbon atom of the C=N bond of compound **6** has a chemical shift at δ 169.1 ppm. The signals of aromatic carbon atoms for compounds **4a**–**e** and **5a**–**e** are in the range δ 112.5–161.0 ppm; *para-*СН_3_О and *para*-CH_3_ groups of the compounds **4****а**, **5****а** and **4b**, **5b**, respectively, give signals at δ 55.4 ppm and δ 21.3 ppm.

As we have proved in this article, 1,2,4-oxadiazoles **4a**–**e** are recorded spectroscopically (FT-IR and NMR spectral data), and they are the initial ones during hydrolysis to pyrazolinium compounds. The Boulton–Katritsky rearrangement mechanism **4a**–**e**→**5a**–**e**, and **4a**–**e**→**6** can be represented as a sequence of protonation, proton transfer and nucleophilic attack steps, representing hydrolysis during the reaction of 1,2,4-oxadiazoles **4a**–**e** with water and wet HCl in the same way as we indicated for piperidine derivatives [9].

All these data demonstrate that the biological activity of compounds under discussion should be associated with their spiropirazolinium form.

### 2.2. In Vitro Antitubercular Screening of 2-amino-8-oxa-1,5-diazaspiro[4.5]dec-1-en-5-ium Benzoates and Chloride (***5a**–**e**, **6***)

In vitro antitubercular bacteriostatic activity of spiropyrazolinium compounds **5a**–**e** and **6** on drug-sensitive (DS) and multidrug-resistant (MDR) of *M. tuberculosis* (MTB) strains was studied using the method of serial dilution on the liquid Shkolnikova medium (Table 2).

A number of compounds **5a**–**e** on DS and MDR MTB strains exhibits an average antitubercular activity of 10–100 μg/mL. Moreover, an improvement in activity is observed with a decrease in MIC to 10 μg/mL (**5b**); 20 μg/mL (**5c**) and 50 μg/mL (**5a**) on the DS MTB strains and up to 50 μg/mL on the MDR MTB strains for the compounds **5a**–**c** containing donor substituents in the phenyl ring or with an unsubstituted phenyl ring**.** Spiropyrazolylammonium chloride **6** demonstrates high in vitro antitubercular activity equal to the activity of the basic antitubercular drug of the first-row rifampicin: on the DS strain as low as 1 μg/mL; on the wild MDR strain—2 μg/mL.

To rationalize the results of the antitubercular activity of 2-amino-8-oxa-1,5-diazaspiro[4.5]dec-1-en-5-ium salts, X-ray diffraction studies and molecular docking studies were carried out.

### 2.3. X-ray Diffraction

Molecular structures of compounds **5c**–**e** and **6** are given in Figure 1. Asymmetric units of **5c**, **5e** and **6** contain one water molecule besides the target ions, and the asymmetric unit of **5d** contains two cations and two symmetrically independent anions. All hydrogen atoms could be located on residual density maps; thus, it was confirmed that the structures are salts with deprotonated carboxylic acids. The six-membered oxo-containing cycles adopt the chair conformation, and the five-membered aza-containing cycles realize the envelope conformation. C(1) carbon atom deviates from the meanplane of N1=N2-C3-C2 atoms at 0.40(1)–0.49(1) Å. Although the cation is rather rigid, it can realize two conformational isomers depending on the shift of the C(1) atom from the meanplane of the rest atoms of the five-membered ring. In crystals of **5c**, **5e** and **6**, these isomers are related with each other by an inversion center, and in acentric crystal **5d**, two symmetrically independent cations realize different conformations (Figure 2a). Nevertheless, for cations in **5c**–**e** and **6** with similar conformations, the mean atomic deviation doesn′t exceed 0.5Å (Figure 2b). The positive charge on quaternary ammonium atom causes elongation of N(1)-N(2) and N(1)-C(1) bonds as it was previously demonstrated for 5-aryl-3-[2-(piperidin-1-yl)ethyl]-1,2,4-oxadiazoles [9], 2-amino-8-thia-1-aza-5-azoniaspiro[4.5]dec-1-ene [7], 1-(*tert*-butyl)-4,5-dihydro-1*H*-pyrazol-1-ium [19] and 1,1,3-trimethyl-Δ^2^-pyrazolinium [20] analogs.

Crystal packing of a molecule can give valuable information about the most abundant intermolecular interactions of a polytopic molecule. The discussed cations can act as donors of two hydrogen bonds and, theoretically, acceptors of three H-bonds (through O(1), N(2) and N(3) atoms). However, the Full Interaction Maps tool [21,22] implemented within the Mercury 2020.1 package [23] undoubtedly indicates the “inertness” of this cation as an acceptor of H-bonds (Figure 3). A monocarboxylate can only be an acceptor of 2–4 H-bonds. Thus, in the absence of water molecules, the cations and anions form simple H-bonded chains (Figure 4) [9]. In **5d**, these chains are further connected through halogen C-Br..O interactions (Figure 4a). In crystals of **5c**, **5e,** and **6,** water molecules act as linkers between the anion and cation and additionally bind two chains through O-H...O(anion) or O-H...Cl^−^ interactions (Figure 5) so that topologically identical H-bonded chains of the ladder-type are formed.

To sum up, X-ray diffraction data demonstrated that molecular docking studies should be carried out for two molecular conformations and that for stable ligand–receptor complexes, the most likely hydrogen bond occurs through the NH_2_ group, while acceptor atoms of heterocycles less readily take part in the hydrogen bonds.

### 2.4. Molecular Docking Studies

Our results indicate that compounds **4a**–**e** at neutral pH in aqueous solutions undergo transformation to **5a**–**e**, and antitubercular activity demonstrated by the latter compounds can be referred to either anion of benzoic acids, or the cation, or a mixture of an anion and cation. Some benzoic acids and their anions previously showed activity against *M. tuberculosis* in vitro [24,25,26]. However, it was assumed that benzoic acids do not have a specific cellular target for *M. tuberculosis* besides their general effect on disrupting the membrane function [25], which is supported by X-ray data of some benzoic acid derivatives with receptors, where ligands are situated on the protein surface [see, for example, PDB id 5TJZ, 5TJY, 2HRG]. Moreover, the activity of **5a**–**e** decreased in comparison with monocarboxylate-free salt **6** allows suggesting that the antitubercular activity of these salts should be referred to as the cation mainly.

As we mentioned above, the cation has a rigid conformation with spirocyclic motifs known to be present in some drugs [27,28]. Thus, the mutual disposition of donor, acceptor and hydrophobic fragments of this molecule can be compared with previously X-rayed drugs and ligand–receptor complexes. We used CSD-Crossminer 2020.1 package to search in the Protein Data Bank [29] ligand–receptor complexes where a ligand contains an (*i*) planar heterocycle with (*ii*) donor amine group involved in two H-bonds and (*iii*) neighboring ring, and a receptor was obtained from *M. tuberculosis*. The search gave three hits with PDB id codes 1NBU, 4FOG, and 4XT4, and R.M.S.D. of pharmacophore model from experimental ligand–receptor complexes of 0.398–0.717 Å. For all these hits,2-amino-8-oxa-1,5-diazaspiro[4.5]dec-1-ene-5-ammonium emulates the pterin moiety, known as a precursor of tetrahydrofolate synthesis, a coenzyme that acts as a carrier for one-carbon units in the biosynthesis of thymidylate, purine nucleotides, and some amino acids [30]. Receptors for these complexes include dihydroneopterin aldolase, thymidylate synthase or oxidoreductase Rv2671. The second search was carried out for structural analogs of spiro[4.5]decane in complexes with receptors obtained from *M. tuberculosis*. The second search gave two hits (PDB codes 5ICJ and 5N7O) of *M. tuberculosis* regulatory protein with 1-oxa-2,8-diazaspiro[4.5]dec-2-en-8-yl derivatives. These receptors were taken as targets for molecular docking calculations together with more typical for theoretical antitubercular screening UDP-galactopyranose mutase.

For all cases, both stereoisomers of the cation were docked independently. Docking without constraints does not give any ligand: receptor complexes; thus, docking was performed with H-bond constraints similar to that in the initial ligand–receptor complexes. Sterical clashes in the binding pocket of thymidylate synthase and UDP-galactopyranose mutase (PRB id codes, respectively, 4FOG and 4RPJ) do not allow the cation to realize appropriate H-bonding; thus, these receptors were excluded from consideration. For 1NBU and 4XT4, we succeeded in overcoming some steric clashes when flexible residues were allowed to rotate freely along with single bonds. Energies of interactions between the cations and three receptors are listed in Table 3; the closest environment of the cation is schematically represented in Figure 6.

Overall, energies of ligand–receptor interactions vary from −35.84 to −42.84 kcal/mol, with the most prominent contribution from nonpolar C-H...π interactions. These values are close to the value of intermolecular pairwise interactions of the same cation in the crystal of **6** (−35.3 kcal/mol) estimated using UNI potentials [31,32]. The cation within binding pockets is involved in two or three hydrogen bonds, and for dihydroneopterin aldolase and thymidylate synthase complexes, the best solution for molecules type A and type B differs in H-bonding patterns. For 1NBU, the best docking solution for molecule type A includes three hydrogen bonds (two N-H...O interactions of amine and oxygen atoms of Tyr52 and Glu74 residues, additionally supported with N-H...N bonds between amide of Asn44 and heterocycle), while the solution for type B molecules includes two H-bonds (N-H...O between amine and Tyr52 and N-H...O between ammonium of Lys99 and the oxygen atom of the cation). The latter solution with the poorest system of H-bonds also has the highest energy of intermolecular interactions among the six solutions under discussion. For 4XT4, both isomers of cation form similar H-bonds with amide group of Asn44 and carboxylate group of Asp67, and interact not only with receptor but also cofactor. The nature of ligand-cofactor interactions differs for two isomers, as well as the system of H-bonds. Only in 5ICJ, both isomers realize nearly similar ligand–receptor interactions through two H-bonds between hydrogen atoms of NH_2_ group, and OH group of Ser134 and COO group of Asp168 and numerous hydrophobic interactions with indolyl fragment of Trp100 additionally supported by hydrophobic interactions with Thr138 and Leu167. To summarize, based on molecular docking calculations, the cation is able to bind with *M. tuberculosis* dihydroneopterin aldolase, thymidylate synthase and regulatory proteins of transcription. Only in the latter case binding energy, the system of H-bonds and contributions of various types of interactions are independent on stereoisomer of the cation. However, other studies are needed to reveal the mechanism of antitubercular activity of compounds under discussion.

## 3. Materials and Methods

### 3.1. Synthesis

The reagents were purchased from different chemical suppliers and were purified before use. FT-IR spectra were obtained on a Thermo Scientific Nicolet 5700 FTIR instrument (Waltham, MA USA) in KBr pellets. ^1^H- and ^13^C-NMR spectra were acquired on a Bruker Avance III 500 MHz NMR spectrometer (500 and 126 MHz, respectively) (Bruker, BioSpin GMBH, Rheinstetten, Germany). The signals of DMSO-*d*6 were used as the internal reference for ^1^H-NMR (2.50 ppm) and ^13^C-NMR (39.5 ppm) spectra. Elemental analysis was carried out on a CE440 elemental analyzer (Exeter Analytical, Inc., Shanghai, China). Melting points were determined in glass capillaries on a PTP(M) apparatus (Khimlabpribor, Klin, Russia). The reaction progress and purity of the obtained products were controlled using Sorbfil (Sorbpolymer, Krasnodar, Russia) TLC plates coated with CTX-1A silica gel, grain size 5–17 μm, containing UV-254 indicator. The eluent for TLC analysis was a mixture of benzene–EtOH, 1:3. The solvents for synthesis, recrystallization, and TLC analysis (ethanol, 2-PrOH, benzene, DMF, acetone) were purified according to the standard techniques.

#### 3.1.1. A General Procedure for the Synthesis of 5-Aryl-3-[2-(morpholin-1-yl)ethyl]-1,2,4-oxadiazoles (4**a**–**e**)

A solution of О-aroyl-(β-morpholin-1-yl)propioamidoximes (**3a**–**e**) in dry DMF in a ratio of 1 g:5 mL, respectively, was heated on an oil bath at 70 °C for different times characteristic for compounds **3a**–**e**: 3.5 h (**3a**, **3****b**), 2.5 h (**3c**), 2 h (**3d**), 1.5 h (**3e**) with TLC control (eluent was mixture benzene—EtOH, 1:3). The reaction mixture was evaporated to dryness under an oil pump vacuum at 50 °C/1 mm Hg. The organic residue was treated with dry acetone at a ratio of 1 g of parent compounds (**3a**–**e**): 5 mL of dry acetone. The obtained crude precipitates of the compounds **4a**–**e** were recrystallized from 2-PrOH.

5*-(4-Methoxyphenyl)-3-[2-(morpholin-1-yl)ethyl]-1,2,4-oxadiazole (***4a***).* Starting from (4.93 g, 0.016 mol) of **3a** in 25 mL of dry DMF to the resulting 2.89 g (62%), colorless solid **4a**, m.p. 230 °C, *R*_f_ 0.71. IR (KBr, cm^–1^): 1670 (C=N), 1597 (C=N), 1553 (C=C), 1362 (C–O). ^1^H-NMR (500 MHz, DMSO-*d*_6_): 3.17 (t, *J* = 7.0 Hz, 2H, CH_2_CH_2_N), 3.40 [m, 2H, N(CH eq)_2_] and 3.93 [m, 6H (2H, N(CH ax)_2_ and 4H, O(CH_2_)_2_)], 3.65 (t, *J* = 7.0 Hz, 2H, CH_2_CH_2_N), 3.73 (s, 3H, *p*-CH_3_O), 6.75 (d, *J* = 8.0 Hz, 2H, *o*-H Ar), 7.75 (d, *J* = 8.0 Hz, 2H, *m*-H Ar). ^13^C-NMR (126 MHz, DMSO-*d*_6_): 31.4, 55.4, 62.0, 62.4, 112.5, 130.9, 136.0, 160.0, 168.9, 169.3. Anal. Calcd for C_15_H_19_N_3_O_3_ (289,33): C, 62.27; H, 6.62. Found: C, 62.60; H, 7.02.

*5-(4-Tolyl)-3-[2-(morpholin-1-yl)ethyl]-1,2,4-oxadiazole* (**4b**). Starting from (2.81 g, 0.0096 mol) of **3a** in 15 mL of dry DMF to the resulting 1.84 g (70%), colorless solid **4b**, m.p. 220 °C, *R*_f_ 0.62. IR (KBr, cm^–1^): 1648 (C=N), 1593 (C=N), 1553 (C=C), 1376 (C–O). ^1^H-NMR (500 MHz, DMSO-*d*_6_):2.27 (s, 3H, *p*-CH_3_), 3.15 (t, *J* = 7.0 Hz, 2H, CH_2_CH_2_N), 3.40 [m, 2H, N(CHeq)_2_] and 3.91 [6H, m: 2H, N(CH ax)_2_ and 4H, O(CH_2_)_2_], 3.66 (t, *J* = 7.0 Hz, 2H, CH_2_CH_2_N), 7.01 (d, *J* = 8.0 Hz, 2H, *o*-H Ar), 7.70 (d, *J* = 8.0 Hz, 2H, *m*-H Ar). ^13^C-NMR (126 MHz, DMSO-*d*_6_): 21.3, 31.4, 62.1, 62.4, 63.2, 112.5, 130.9, 136.0, 160.0, 169.0, 169.3. Anal. Calcd for C_15_H_19_N_3_O_2_ (273,33): C, 65.91; H, 7.01. Found: C, 65.50; H 7.22.

*5-Phenyl-3-[2-(morpholin-1-yl)ethyl]-1,2,4-oxadiazole* (**4c**). Starting from (1.00 g, 0.0036 mol) of **3с** in 5 mL of dry DMF to the resulting 0.86 g (92%) colorless solid **4c**, m.p. 216 °C, *R*_f_ 0.66. IR (KBr, cm^–1^): 1650 (C=N), 1596 (C=N), 1559 (C=C), 1381 (C–O). ^1^H-NMR (500 MHz, DMSO-*d*_6_): 3.18 (t, *J* = 7.0 Hz, 2H, CH_2_CH_2_N), 3.40 [2H, m, N(CHeq)_2_] and 3.93 [6H, m: 2H, N(CHax)_2_ and 4H, O(CH_2_)_2_], 3.65 (2H, t, *J* = 7.0 Hz, CH_2_CH_2_N), 7.23–7.83 (m, 5H, C_6_H_5_). ^13^C-NMR (126 MHz, DMSO-*d*_6_): 31.4, 62.1, 62.4, 63.3, 127.5, 127.8, 128.9, 129.1, 129.4, 167.0, 169.2. Anal. Calcd for C_14_H_17_N_3_O_2_ (259,30):C, 64.85; H, 6.61. Found: C, 65.30; H, 7.02.

*5-(4-Bromophenyl)-3-[2-(morpholin-1-yl)ethyl]-1,2,4-oxadiazole* (**4d**). Starting from (1.44 g, 0.004 mol) of **3d** in 8 mL of dry DMF to the resulting 1.12 g (83%) colorless solid **4d**, m.p. 224 °C, *R*_f_ 0.67. IR (KBr, cm^–1^): 1650 (C=N), 1596 (C=N), 1559 (C=C), 1381 (C–O). ^1^H-NMR (500 MHz, DMSO-*d*_6_): 3.17 (t, *J* = 7.0 Hz, 2H, CH_2_CH_2_N), 3.39 [m, 2H, N(CHeq)_2_] and 3.93 [m, 6H: 2H, N(CH ax)_2_ and 4H, O(CH_2_)_2_], 3.65 (2H, t, *J* = 7.0 Hz, CH_2_CH_2_N), 7.40 (d, *J* = 7.5 Hz, 2H, *o*-H Ar), 7.74 (d, *J* = 7.5 Hz, 2H, *m*-H Ar). ^13^C-NMR (126 MHz, DMSO-*d*_6_): 31.4, 62.1, 62.4, 63.3, 122.7, 130.2, 131.6, 141.5, 168.3, 169.5. Anal. Calcd for C_14_H_16_BrN_3_O_2_ (338.20): C, 49.72; H, 4.77. Found,%: C, 49.29; H, 4.97.

*5-(3-Chlorophenyl)-3-[2-(morpholin-1-yl)ethyl]-1,2,4-oxadiazole (***4e***).* Starting from (2.70 g, 0.0087 mol) of **3e** in 15 mL of dry DMF to the resulting 1.43 g (56%) colorless solid **4e**, m.p. 190 °C, *R*_f_0.62. IR (KBr, cm^–1^): 1680 (C=N), 1596 (C=N), 1557 (C=C), 1377 (C–O). ^1^H-NMR (500 MHz, DMSO-*d*_6_): 3.15 (t, *J* = 7.0 Hz, 2H, CH_2_CH_2_N), 3.40 [m, 2H, N(CHeq)_2_] and 3.91 [m, 6H: 2H, N(CHax)_2_ and 4H, O(CH_2_)_2_], 3.65 (t, *J* = 7.0 Hz, 2H, CH_2_CH_2_N), 7.25–7.78 (m, 4H, 3-Cl-C_6_H_4_). ^13^C-NMR (126 MHz, DMSO-*d*_6_): 31.4, 62.1, 62.4, 63.2, 127.7, 127.8, 129.2, 132.4, 144.5, 162.1, 169.2. Anal. Calcd for C_14_H_16_ClN_3_O_2_ (293.75): C, 57.24; H, 5.49. Found,%: C, 57.55; H, 5.92.

#### 3.1.2. A General Procedure of the Formation of 2-Amino-8-oxa-1,5-diazaspiro[4.5]dec-1-ene-5-ammonium benzoates (**5a**–**e**)

To the solutions of 5-(3,4-substituted phenyl)-3-[2-(morpholin-1-yl)ethyl]-1,2,4-oxadiazoles (**4a**–**e**) in DMF in a ratio of 0.5 g: 10 mL, respectively, two equivalents of H_2_O was added dropwise. The reaction solutions were heated at 70 °C with TLC control during the time typical for groups of compounds **4a**–**e**: **4a**, **b** (40 h), **4c**–**e** (25 h).After the disappearance of the 1,2,4-oxadiazoles **4a**–**e** spots on the silufol plate and the appearance of products **5a**–**e** spots, the solvent was evaporated to dryness in an oil vacuum pump. Acetone was added to the viscous residues of the reaction mixtures **5a**–**e** at the rate of 0.5 g of the starting compounds **4a**–**e**: 15 mL of acetone. The formed precipitates of 2-amino-8-oxa-1,5-diazaspiro[4.5]dec-1-ene-5-ammonium 3,4-substituted benzoates (**5a**–**e**) were filtered off and dried at room temperature.

*2-Amino-8-oxa-1,5-diazaspiro[4.5]dec-1-ene-5-ammonium 4-methoxybenzoate hydrate* (**5a**). Starting from 0.5 g (0.0017 mol) of compound **4a** in 10 mL of DMF and 0.062 mL (0.0034 mol) H_2_O to the resulting 0.35 g (63%) colorless solid **5a**, m.p. 235 °C, *R*_f_ 0.75. IR (KBr, cm^–1^): 1657 (C=N), 1594 (C=C), 1548 (νCOO*^–^* as), and 1420 (νCOO*^–^* sy), 3309, 3414, 3459 [νN(-H)_2_]. ^1^H-NMR (500 MHz, DMSO-*d*_6_): 2.27 (s, 3H, *p*-CH_3_O), 3.17 [t, *J* = 7.0 Hz, 2H, CH_2_CH_2_N(+)], 3.41 [m, 2H, N(CHeq)_2_] and 3.92 [m, 6H: 2H, N(CHax)_2_ and 4H, O(CH_2_)_2_], 3.65 [t, *J* = 7.0 Hz, 2H, CH_2_CH_2_N(+)], 6.78 (d, *J* = 8.0 Hz, 2H, o-H Ar), 7.55 and 7.60 (s, 2H, NH_2_), 7.78 (d, *J* = 8.0 Hz, 2H, *m*-H Ar). ^13^C-NMR (126 MHz, DMSO-*d*_6_): 31.4, 62.4, 63.2, 112.5, 130.9, 134.8, 159.9, 144.5, 168.8, 169.2. Anal. Calcd for C_15_H_23_N_3_O_5_ (325,36): C, 55.37; H, 7.13. Found,%: C, 55.79; H, 7.57.

*2-Amino-8-oxa-1,5-diazaspiro[4.5]dec-1-ene-5-ammonium 4-methylbenzoate hydrate* (**5b**). Starting from 0.5 g (0.0018 mol) of the compound **4b** in 10 mL of DMF and 0.066 mL (0.0036 mol) H_2_O to the resulting 0.35 g (63%) colorless solid **5b**, m.p. 248 °C, *R*_f_ 0.80. IR (KBr, cm^–1^): 1655 (C=N), 1597 (C=C), 1550 (COO*^–^*as), and 1442 (COO*^–^* sy), 3157, 3312, 3457 [N(-H)_2_]. ^1^H-NMR (500 MHz, DMSO-*d*_6_): 3.15 [t, *J* = 7.0 Hz, 2H, CH_2_CH_2_N(+)], 3.41 [m, 2H, N(CHeq)_2_] and 3.93 [m, 6H: 2H, N(CHax)_2_ and 4H, O(CH_2_)_2_], 3.64 [t, *J* = 7.0 Hz, 2H, CH_2_CH_2_N(+)], 3.73 (s, 3H, *p*-CH_3_), 7.02 (d, *J* = 8.0 Hz, 2H, *o-*H Ar), 7.54 (s, 2H, NH_2_), 7.69 (d, *J* = 8.0 Hz, 2H, *m-*H Ar). ^13^C-NMR (126 MHz, DMSO-*d*_6_): 21.3, 31.38, 62.1, 62.4, 63.2, 127.9, 128.3, 129.4, 139.7, 168.8, 169.2. Anal. Calcd for C_15_H_23_N_3_O_4_. (309,36): C, 58.24; H, 7.49. Found: C, 58.29; H, 7.59.

*2-Amino-8-oxa-1,5-diazaspiro[4.5]dec-1-ene-5-ammonium benzoate hydrate* (**5c**). Starting from 0.5 g (0.0019 mol) of **4c** in 10 mL of DMF and 0.07 mL (0.0038 mol) H_2_O to the resulting 0.43 g (77%) colorless solid **5c**, m.p. 220 °C, *R*_f_ 0.75. IR (KBr, cm^–1^):1658 (C=N), 1596 (C=C), 1556 (COO*^–^* as), and 1426 (COO*^–^* sy), 3152, 3311, 3456 [N(-H)_2_]. ^1^H-NMR (500 MHz, DMSO-*d*_6_): 3.19 [t, *J* = 7.0 Hz, 2H, CH_2_CH_2_N(+)], 3.39 [m, 2H, N(CHeq)_2_] and 3.93 [m, 6H: 2H, N(CHax)_2_ and 4H, O(CH_2_)_2_], 3.65 [t, *J* = 7.0 Hz, 2H, CH_2_CH_2_N(+)], 7.22–7.80 (m, 5H, C_6_H_5_), 7.55 and 7.64 (s, 2H, NH_2_). ^13^C-NMR (126 MHz, DMSO-*d*_6_): 31.4, 61.3, 62.4, 63.2, 127.26, 128.3, 129.4, 142.4, 168.7, 169.2. Anal. Calcd for C_14_H_21_N_3_O_4_ (295,33): C, 56.94; H, 7.17. Found: C, 56.79; H, 7.50.

*2-Amino-8-oxa-1,5-diazaspiro[4.5]dec-1-ene-5-ammonium 4-bromobenzoate* (**5d**). Starting from 0.5 g (0.0015 mol) of **4d** in 10 mL of DMF and 0.05 mL (0.0028 mol) H_2_O to the resulting 0.43 g (80%) colorless solid **5d**, m.p. 240 °C, *R*_f_ 0.77. IR (KBr, cm^–1^): 1657 (C=N), 1591 (C=C), 1545 (COO*^–^* as_)_, and 1423 (COO*^–^*
*s*y), 3302, 3411, 3485 [N(-H)_2_]. ^1^H-NMR (500 MHz, DMSO-*d*_6_): 3.15 [t, *J* = 7.0 Hz, 2H, CH_2_CH_2_N(+)], 3.40 [m, 2H, N(CHeq)_2_] and 3.92 [m, 6H: 2H, N(CHax)_2_ and 4H, O(CH_2_)_2_], 3.65 [t, *J* = 7.0 Hz, 2H, CH_2_CH_2_N(+)], 7.39 (d, *J* = 7.5 Hz, 2H, *o-*H Ar), 7.55 (s, 2H, NH_2_), 7.73 (d, *J* = 7.5 Hz, 2H, *m-*H Ar). ^13^C-NMR (126 MHz, DMSO-*d*_6_): 31.4, 62.1, 62.4, 63.2, 121.9, 130.1, 131.6, 141.8, 167.4, 169.2. Anal. Calcd for C_14_H_18_BrN_3_O_3_ (356,22): C, 47.20; H, 5.09. Found: C, 47.49; H, 4.97.

*2-Amino-8-oxa-1,5-diazaspiro[4.5]dec-1-ene-5-ammonium 3-chlorobenzoate hydrate* (**5e**). Starting from 0.5 g (0.0017 mol) **4e** in 10 mL of DMF and 0.06 mL (0.034 mol) H_2_O to the resulting 0.24 g (43%) colorless solid **5e**, m.p. 200 °C, *R*_f_ 0.70. IR (KBr, cm^–1^): 1657 (C=N), 1594 (C=C), 1557 (COO*^–^* as), and 1420 (COO*^–^* sy), 3155, 3309, 3457 [N(-H)_2_]. ^1^H-NMR (500 MHz, DMSO-*d*_6_): 3.16 [t, *J* = 7.0 Hz, 2H, CH_2_CH_2_N(+)], 3.40 [m, 2H, N(CH eq)_2_] and 3.93 [6H, m: 2H, N(CHax)_2_ and 4H, O(CH_2_)_2_], 3.66 [t, *J* = 7.0 Hz, 2H, CH_2_CH_2_N(+)], 7.25–7.78 (m, 4H, 3-Cl-C_6_H_4_), 7.57 and 7.61 (s, 2H, NH_2_). ^13^C-NMR (126 MHz, DMSO-*d*_6_): 31.4, 62.1, 62.5, 63.2, 127.9, 128.2, 129.2, 129.3, 132.4, 144.7, 167.0, 169.2. Anal. Calcd for C_14_H_20_ClN_3_O_4_ (329,78): C, 50.99; H, 6.11. Found: C, 50.52; H, 6.47.

#### 3.1.3. A General Method of the Formation of 2-Amino-8-oxa-1,5-diazaspiro[4.5]dec-1-ene-5-ammonium chloride (**6**) and Substituted Benzoic Acids

To a solution of 0.2 g (0.0007 mol) of 5-(3,4-substituted phenyl)-3-[2-(morpholin-1-yl)ethyl]- 1,2,4-oxadiazoles (**4a**–**e**) in 10 mL ethanol a HCl solution in ether was added dropwise to pH 2. A white precipitate of 2-amino-1,5-diazaspiro[4.5]dec-1-en-5-ium chloride hydrate (**6**) in all cases was formed at once in 75–82% yields and collected by filtration. 3,4-Substituted benzoic acids were precipitated during the evaporation of mother liquors obtained after the filtration of chloride monohydrate **6**. All characteristics of substituted benzoic acids corresponded to the reference data.

*2-Amino-8-oxa-1,5-diazaspiro[4.5]dec-1-ene-5-ammonium chloride hydrate* (**6**)—white opaque powder, m.p. 257–260 °C, *R*_f_ 0.21. IR (KBr, cm^–1^): 1639 (νC=N), [1615, δN(–H)_2_], 1362 (νC–O), 3158, 3310, 3457 [νN(-H)_2_]. ^1^H-NMR (500 MHz, DMSO-*d*_6_): 3.16 [t, *J* = 7.0 Hz, 2H, CH_2_CH_2_N(+)], 3.40 [m, 2H, N(CH eq)_2_] and 3.92 [m, 6H: 2H, N(CH ax)*_2_* and 4H, O(CH_2_)_2_], 3.68 [t, *J* = 7.0 Hz, 2H, CH_2_CH_2_N(+)], 7.51 (s, 2H, NH_2_). ^13^C-NMR spectrum, δ, ppm: 31.4, 31,5, 62.1, 62.4, 63.3, 169.2. Anal. Calcd for C_7_H_16_N_3_O_2_ (209.67): C, 40.10, H 7.69. Found: C, 40.29; H, 7.97.

### 3.2. Single-Crystal X-ray Diffraction

The X-ray diffraction data of **5c**–**e**, **6** were collected at 120 K on a Bruker Apex II diffractometer (Bruker AXS, Inc., Madison, WI, USA) equipped with an Oxford Cryostream cooling unit and a graphite monochromated Mo anode (λ = 0.71073 Å). Crystal structures were solved using SHELXT [33] program and refined with SHELXL [34] using OLEX2 software [35]. The structures were refined by full-matrix least-squares procedure against F^2^. Non-hydrogen atoms were refined anisotropically. The H(C) positions were calculated, the H(N) and H(O) atoms were located on difference Fourier maps and refined using the riding model. Experimental details and crystal parameters are given in Appendix A (Electronic Supporting Information).

### 3.3. Molecular Docking Studies

The molecular docking studies were performed following the previously described protocol [36,37] and processed with GOLD software (version 2020.2.0, Cambridge Crystallographic Data Center, Cambridge, UK).

The three-dimensional (3D) crystal structures of *M. tuberculosis* regulatory protein (PDB ID: 5ICJ), 7,8-dihydroneopterin aldolase complexed with 2-amino-6-(hydroxymethyl)pteridin-4(3*H*)-one (PH2; PDB ID: 1NBO), thymidylate synthase in complex with 5-fluoro DUMP (UFP; PDB ID: 4FOG), UDP-galactopyranose mutase from Mtb docked with UDP (UGM; PDB ID: 4RPJ), and Rv2671 protein in complex with dihydropteridine (44W; PDB ID: 4XT4) were obtained from the RCSB Protein Data Bank [29], while the 3D-structure of cation was taken from X-ray diffraction data of **6** (two stereoisomers were analyzed independently).

To validate the molecular docking outcomes, 4,4,4-trifluoro-1-(3-phenyl-1-oxa-2,8-diazaspiro[4.5]dec-2-en-8-yl)butan-1-one, PH2, UFP, UGM, and 44W were removed from their receptors and re-docked back into their receptors. The docking results were expressed as binding energy values (kcal/mol) of ligand–receptor complexes; these are based on hydrogen bond, hydrophobic, and electrostatic interactions. All required docking settings, including the preparation of mol2 files for the receptors and ligands, determination of binding sites, the protonation state, calculations, and the overall charges, were established as hitherto described [38].

## 4. Conclusions

5-Substituted phenyl-3-(2-aminoethyl)-1,2,4-oxadiazoles in the presence of moisture and acids undergo Boulton–Katritsky rearrangement to the salts of spiropyrazolinium compounds—2-amino-8-oxa-1,5-diazaspiro[4.5]dec-1-en-5-ium benzoates and chloride. Hence, biological screening results should be associated with rearranged products and not with the original taken on trials 5-substituted phenyl-3-(2-aminoethyl)-1,2,4-oxadiazoles. A small library of the newly 2-amino-8-oxa-1,5-diazaspiro[4.5]dec-1-en-5-ium benzoates and chloride has been used in the drug design of antitubercular drugs. The tested compounds show moderate to high in vitro antitubercular activity with MIC values of 1–100 µg/mL. The highest activity in 1 µg/mL and 2 µg/mL on DS and MDR of *M. tuberculosis* strains, equal to the activity of the basic antitubercular drug rifampicin, was recorded for 2-amino-8-oxa-1,5-diazaspiro[4.5]dec-1-en-5-ium chloride. 3D molecular structure of the cation extracted from crystal structure was used for molecular docking studies with various *M. tuberculosis* receptors. It was demonstrated that two stereoisomers of the rigid cation form different sets of hydrogen bonds in complexes with dihydroneopterin aldolase or oxidoreductase Rv2671 and similar H-bonds in complex with thymidylate synthase. However, energies of all ligand–receptor complexes vary from −35.8 to −42.8 kcal/mol.

## Data Availability

The data presented in this study are available in this article.

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
