# Peer review of "Boulton-Katritzky Rearrangement of 5-Substituted Phenyl-3-[2-(morpholin-1-yl)ethyl]-1,2,4-oxadiazoles as a Synthetic Path to Spiropyrazoline Benzoates and Chloride with Antitubercular Properties"

_molecules, 2021, doi:10.3390/molecules26040967_

Round 1

Reviewer 1 Report

This is a manuscript in continuation of former article [11].The synthesis is quite similar to the piperidinyl analogue in this reference. The X-ray diffruction and Molecular docking studies are interest.

There are some remarks as specified below:

  • 141-142, 159: Scheme 8 must be changed to Scheme VIII.
  • 364-365: There is one aliphatic C less.
  • 391-392: There is one aromatic C more.
  • 418: There is one aromatic C less.
  • 438-439: The carbon must be checked.

Author Response

w141-142, 159: Thanks to the reviewer for the attentive attitude to the text. According to the authors, one of the references to Scheme VIII on line 141 is superfluous, in the text it has been removed, since there is a reference to Scheme VIII in the next sentence. At present, taking into account the comments of reviewer 3, the number of schemes has been reduced and scheme VIII has the number V.

w 364-365 (357-358 of the revised article): According to the authors, the reviewer is in error, because the number of aliphatic carbon atoms in the compound 4a corresponds to the chemical scheme of the compound with the gross formula C15H19N3O3.

w 391-392 (375-376 of the revised article): The authors agree with the reviewer that the number of carbon atoms in the compound 4 c, and therefore the gross formula given in the article: C15H19N3O3, is incorrect. In this case, you should write the gross formula C14H17N3O2.

w 418 (391-392 of the revised article): The authors checked the structural and gross formulas of compound 4e. They are correct and correspond to those given in the article for the name 5-(4-chlorophenyl)-3-[2-(morpholin-1-yl)ethyl]-1,2,4-oxadiazole (4e) with gross formulas C14H16ClN3O2.  

Anal. Calcd for C14H16ClN3O2 (293.75): C, 57.24; H, 5.49. Found, %: C, 57.55; H, 5.92.

w 438‒439 (411-412 of the revised article): The authors checked the structural and gross formulas of compound 5а. They are correct and correspond to those given in the article for the name of the compound 2-amino-8-oxa-1,5-diazaspiro[4.5]dec-1-en-5-ammonium 4-methoxybenzoate hydrate with the gross formula C15H23N3O5.

Date of answer: 16 Jan 2021                                                                                                          

From the authors: L.A. Kayukova

Reviewer 2 Report

82   towards hydrolysis, we found that they were capableofrearrangingtospiropyrazolinecompounds   No spaces - is it on purpose?

104 of Boulton-Katritzky rearrangement scould be represented as the following scheme (Scheme VI) [12,  what did you mean in this phrase "scould be"

120 Scheme VII. 2-Amino-4,5-dihydrospiropyrazolylammonium formation atarylsulfochlorination of....   or at arylosulfochlorination ?

The text contains minor editing errors, e.g. no spaces:
236 each other by an inversion center, and in acentric crystal5d
241 2-amino-8-thia-1-aza-5-azoniaspiro[4.5]dec-1-ene[9],
242 1-(tert-butyl)-4,5-dihydro-1H-pyrazol-1-ium[22] 447 5bspot, 640 24. Wood, P.A.; Olsson, T.S.G.; Cole, J.C.; Cottrell, S.J.; Feeder, N.;Galek, P.T.A.; Groom, C.R.;Pidcock,
641 E.Evaluation of molecular crystal structures using Full Interaction Maps. CrystEngComm. 2013, 15,

Reviewer 3 Report

The authors reported the Boulton–Katritzky rearrangement of 5-substituted phenyl-3-[2-(morpholin-1-yl)ethyl]-1,2,4-oxadiazoles as a synthetic path to spiropyrazoline benzoates and chloride with antitubercular properties. I recommend to accept this manuscript after minor revision. I did not find the conclusion section and supplementary material. All NMR spectra of synthesized compounds 4, 5 and 6 should be added and properly processed including the integration of each signal (1H NMR) and chemical shifts (both 1H and 13C{1H} NMR) after previous calibration with TMS or deuterated solvent. Additional recommendations and doubts are included to improve this manuscript: 

(1) See abstract. It should be re-structured according to the guideline of this journal. For instance: (a) the following paragraph should be removed “2-aminopropioamidoxime derivatives attract considerable interest as local anesthetics, antitubercular, and antidiabetic agents” (it should be added in the introduction), (b) some paragraphs contain the synthetic procedure (it should be shorten), and finally (c) the most relevant results of antitubercular activity and molecular docking studies should be included.

(2) See keywords. It should be shorten. For instance: 5-substituted phenyl-3-[2-(morpholin-1-yl)ethyl]-1,2,4-oxadiazoles and 2-amino-8-oxa-1,5-diazaspiro[4.5]dec-1-ene-5-ammonium compounds.

(3) See introduction. It looks as a review instead of full paper. It should be re-structured and shorten. It is too long.  

(4) See all schemes. Authors should not use roman numerals to list these schemes.

(5) See page 5, lines 152-153. Authors mention that “This indicates the hydrolysis of 1,2,4-oxadiazoles 4а‒е under the influence of air moisture”. It should be improved to clarify the formation of compounds 5а‒е from 4а‒е. For instance: This indicates that the hydrolysis of 1,2,4-oxadiazoles 4а‒е under the influence of air moisture affording О-aroyl-(β-morpholin-1-yl)propioamidoximes, followed by a Boulton–Katritzky rearrangement to lead unexpected 2-amino-8-oxa-1,5-diazaspiro[4.5]dec-1-en-5-ium salts.

(6) See Scheme 8. The reaction for the formation of compounds 4а‒е from 3а‒е is clearly reversible. The arrow should be changed and also added only one molecule of water.     

(7) See lines 156-168. It is important to unify these paragraphs. The explanation is disorganized.   

(8) See table 1. The substituent (X) should be included. It could be re-structured. For instance, Rf and Mw values are not relevant.

(9) See page 6, lines 178-186. These analysis should be improved. However, the information of the lines 195-208 is very interesting. It should be unified.  

(10) See figures 1 and 2. It should be included in the supplementary material. A properly explanation and organization of NMR information may avoid the inclusion of figures 1 and 2 in the manuscript. However, If you want to include NMR spectra, it should be improved (i.e. resolution and size).  

(11) See table 2. The MIC values in μg/ml should be included instead of symbols like +, ++ and +++. It is important to check articles related to medicinal chemistry (i.e., J. Med. Chem., Eur. J. Med. Chem., Bioorg. Chem., among others) in order to see the presentation of these information. However, the explanation of these results containing MIC values in μg/ml.    

(12) See material and methods. It is important to see a paper recently published in Molecules related to organic synthesis. For instance: (a) The 13C NMR data should be reported with only one instead of two decimals after the point, (b) the deuterated solvent and frequency should be included [i.e. 1H NMR (400 MHz, CDCl3): d =] and/or [i.e. 13C{1H} NMR (101 MHz, CDCl3): d =], (c) the solvent or mixture of solvents for recrystallization process should be included for the corresponding compound, (d) the symbol of the coupling constant (J) should be in italic, and generally (e) the 1H NMR data should be reported like xx.xx (multiplicity, coupling constant, integration, assignation).    

(13) See material and methods. It is not necessary to repeat the procedure of each compound. A general procedure should be included for each family of synthesized compounds. It is important to see a paper recently published in Molecules related to organic synthesis.

(14) See material and methods. See Table 4. It should be included in the supplementary material.

(15) I did not find the supplementary material. All NMR spectra of synthesized compounds 4, 5 and 6 should be added and properly processed including the integration of each signal (1H NMR) and chemical shifts (both 1H and 13C{1H} NMR) after previous calibration with TMS or deuterated solvent.  

 (16) I did not find the conclusion section. It should be included according to the guideline of this journal.                   

Author Response

(1) See abstract. It should be re-structured according to the guideline of this journal. For instance: (a) the following paragraph should be removed “2-aminopropioamidoxime derivatives attract considerable interest as local anesthetics, antitubercular, and antidiabetic agents” (it should be added in the introduction), (b) some paragraphs contain the synthetic procedure (it should be shorten), and finally (c) the most relevant results of antitubercular activity and molecular docking studies should be included.

Answer: The authors revised the abstract in accordance with the necessary requirements (lines 21-38).

Analysis of stability of biologically active compounds requires an accurate determination of their structure. We have found that 5-aryl-3-(2-aminoethyl)-1,2,4-oxadiazoles are generally unstable in the presence of acids and bases and are rearranged into the salts of spiropyrazolinium compounds. Hence, there is a significant probability that it is the rearranged products that should be attributed to biological activity, and not the primary screened 5-aryl-3-(2-aminoethyl)-1,2,4-oxadiazoles. A series of the 2-amino-8-oxa-1,5-diazaspiro[4.5]dec-1-en-5-ium (spiropyrazoline) benzoates and chloride was synthesized by Boulton-Katritzky rearrangement of 5-substituted phenyl-3-[2-(morpholin-1-yl)ethyl]-1,2,4-oxadiazoles, and characterized using FT-IR and NMR spectroscopy and X-ray diffraction. Spiropyrazolylammonium chloride demonstrates in vitro antitubercular activity on DS (drug sensitive) and MDR (multidrug resistant) of MTB (M. tuberculosis) strains (1 and 2 µg/ml, accordingly) equal to activity of the basic antitubercular drug rifampicin; spiropyrazoline benzoates exhibit an average antitubercular activity of 10–100 μg/ml on MTB strains. Molecular docking studies revealed a series of M. tuberculosis receptors with the energies of ligand - receptor complexes (-35.8 ‒ -42.8 kcal/mol) close to the value of intermolecular pairwise interactions of the same cation in the crystal of spiropyrazolylammonium chloride (-35.3 kcal / mol). However, only in complex with transcriptional repressor EthR2 both stereoisomers of the cation realize similar intermolecuar interactions.

 (2) See keywords. It should be shorten. For instance: 5-substituted phenyl-3-[2-(morpholin-1-yl)ethyl]-1,2,4-oxadiazoles and 2-amino-8-oxa-1,5-diazaspiro[4.5]dec-1-ene-5-ammonium compounds.

Answer: In accordance with the requirements of the reviewer, the authors provide a modified list of keywords (lines 39-40).

Keywords: 1,2,4-oxadiazoles; spiropyrazolinium compounds; in vitro antitubercular screening; X-ray diffraction; molecular docking.

(3) See introduction. It looks as a review instead of full paper. It should be re-structured and shorten. It is too long.  

Answer: Dear Reviewer! In connection with your request, the authors have reduced the introduction to 3 literary references and 3 schemes (lines 41-109) and 3 references (lines 534-574).

  1. Introduction

There are pyrazoline-containing compounds that act as active pharmaceutical ingredients of such commercially available drugs as Aminopyrine (Aminophenazone; analgesic and antipyretic), Dipyrone (Metamizol, Noramidopyrine; analgesic), Antipyrine (Benzocaine; non-narcotic analgesic, an antipyretic and anti-rheumatic), Zaleplon (hypnotic and sedative), Celecoxib (Aclarex, Celebrex; anti-inflammatory and antirheumatic drug), Allopurinol (uricostatic agent, xanthine oxidase inhibitor) [1]. Therefore, there is always demand for new molecules, methodologies and improved synthetic approaches to novel pyrazoline derivatives.

Pyrazolines as noticeable practically meaningful nitrogen-containing heterocyclic compounds can be synthesized by a variety of methods. But one of the most popular methods is the Fischer and Knoevenagel synthesis based on the reaction of α,β-unsaturated ketones with phenyl hydrazine in acetic acid under refluxing condition. However, depending on the reactivity of molecules and need of the chemist, they had synthesized the pyrazolines under different solvent media and acidic or basic conditions [2‒4].

Information on pyrazolinium structures with a quaternary nitrogen atom is limited. Thus, two examples of biologically active pyrazolinium salts were found: 3-amino-1-ethyl-1-phenyl-4,5-dihydro-1H-pyrazoliniumiodide (PubChem CID:13585073 structure, [5]) and 3-amino-1,4-dimethyl-1-phenyl-2-pyrazoliniumiodide (PubChemCID: 13064197 structure, [6]) (Scheme I):

Scheme I. 3-Amino-1-phenyl-4,5-dihydro-1H-pyrazolinium iodides.

In some works, we found spiropyrazolinium compounds with a quaternary nitrogen atom, which is common for two heterocycles. When studying the stability of 3-(2-aminoethyl)-5-aryl-1,2,4-oxadiazoles having six-membered cyclic tertiary 2-amino groups towards hydrolysis, we found that they were capable of rearranging to spiropyrazoline benzoates or chlorides [7‒9].

Particularly, upon keeping 3-[2-(4-phenylpiperazin-1-yl)ethyl]-5-phenyl-1,2,4-oxadiazole recrystallized in 2-PrOH under conditions of air moisture access for 9 months for growing single crystals for X-ray structural analysis or by exposure of 3-[2-thiomorpholin-1-yl)ethyl]-5-aryl-1,2,4-oxadiazoles in ethanol with ethereal HCl solution the above mentioned 1,2,4-oxadiazoles underwent the rearrangement to spiropyrazolinium benzoates or chlorides (Scheme II) [7, 8].

Also at targeted exposure on 3-[2-(piperidin-1-yl)ethyl]-5-aryl-1,2,4-oxadiazoles with: (i) water, (ii) water in DMF or (iii) ethereal HCl they underwent rearrangement with the formation 2-amino-1,5-diazaspiro[4.5]dec-1-en-5-ium benzoates or chlorides (Scheme V). In the latter case, along with the formation of spiropyrazolinium chloride hydrate from 3-[2-(piperidin-1-yl)ethyl]-5-(3-Cl-phenyl)-1,2,4-oxadiazole hydrochlorides of starting 1,2,4-oxadiazoles were obtained as secondary products [9].

Scheme II. Behavior of 3-(2-aminoethyl)-5-aryl-1,2,4-oxadiazoles in the conditions of exposure to  

H2O and HCl.

These facts are consistent with the known for 3,5-substituted 1,2,4-oxadiazoles with a saturated side chain spontaneous thermally induced monomolecular Boulton-Katritzky rearrangement and provided the first examples of spirocompound formation through such reaction. In general, a variety of Boulton-Katritzky rearrangement scould be represented as the following scheme (Scheme III) [10, 11]:

Scheme III. Mononuclear heterocyclic Boulton-Katritzky rearrangement of 3,5-substituted 1,2,4-oxadiazoles with saturated side chains.

In addition, spiropyrazolinium structures ‒ 2-amino-8-oxa-1,5-diazaspiro[4.5]dec-1-ene-5-ammonium arylsulfonates are formed at  arylsulfochlorination of β-aminopropioamidoximes (Scheme IV) [12].

Scheme IV. 2-Amino-4,5-dihydrospiropyrazolylammonium formation at arylsulfochlorination of β-aminopropioamidoximes.

The practical interest in the class of β-aminopropioamidoxime derivatives is supported by their pronounced local anesthetic, antitubercular, and antidiabetic activities [13‒16].

Herein we report on the stability of 5-aryl-3-[β-(morpholin-1-yl)ethyl]-1,2,4-oxadiazoles towards hydrolysis at: (i) DMF with the two equivalent amount of water when heated to 60‒70 °C; (ii) alcohol / ethereal HCl mixture. A number of previously unknown spiropyrazolinium salts were obtained and characterized using FT-IR and NMR spectroscopy and X-ray diffraction. In vitro antitubercular screening of spiropyrazoline benzoates and chloride was carried out and their molecular docking was performed. It was shown that the hydrolysis of 1,2,4-oxadiazoles with a 3-morpholinoethyl substituent leads to spiropyrazoline compounds within 25‒40 h. Acid hydrolysis of 1,2,4-oxadiazoles occurs immediately after reagents adding. In vitro antitubercular screening of benzoates and chloride of spiropyrazoline drug susceptible and multidrug resistant strains M. tuberculosis revealed compounds with significant activity; and the results are in accordance with molecular docking studies.

References

  1. List of drugs – Wikipedia.Available online: https://en.wikipedia.org/wiki/List_of_drugs (accessed on 12December2020).
  2. Fischer, E.; Knovenagel, O. Ueber die Verbindungen des Phenylhydrazinsmit Acroleïn, Mesityloxydund Allylbromid. Ann. Chem., 1887, 239, 194‒206. DOI: 10.1002/jlac.18872390205
  3. Santos, C.M.M.; Silva, V.L.M.; Silva, A.M.S. Synthesis of Chromone-Related Pyrazole Compounds. Molecules, 2017, 22, 1665. DOI: 10.3390/molecules22101665
  4. Sharma, S.; Kaur, S.; Bansal, T.; Gaba, J. Review on Synthesis of Bioactive Pyrazoline Derivatives. Sci. Trans.,2014, 3, 861‒875. DOI: 10.7598/cst2014.796
  5. 1-H Pyrazolium at National Library of Medicine.Available online: https://pubchem.ncbi.nlm.nih.gov/compound/13585073(accessed on 12 December2020).
  6. 3-Amino-1,4-dimethyl-1-phenyl-2-pyrazolinium iodide at National Library of Medicine. Available online: https://pubchem.ncbi.nlm.nih.gov/compound/13064197(accessed on 12 December2020).
  7. Kayukova, L.A.; Orazbaeva, M.A.; Gapparova, G.I.; Beketov, K.M.; Espenbetov, A.A.; Faskhutdinov, M.F.;Tashkhodjaev, B.T. Rapid acid hydrolysis of 5-aryl-3-(b-thiomorpholinoethyl)-1,2,4-oxadiazoles. Heterocycl. Compd. 2010, 46, 879‒886. DOI: 10.1007/s10593-010-0597-8
  8. Kayukova, L. A.; Beketov, K. M.; Baitursynova, G. P. In: International Сonference ''Catalysis in Organic Synthesis'', September 15–20, 2012, Moscow, Russia, 2012, Abstr. No. 122, p. 214.
  9. Kayukova, A.; Uzakova, A.B.; Vologzhanina, A.V.; Akatan, K.; Shaymardan, E.; Kabdrakhmanova,  S.K. Rapid Boulton–Katritzky rearrangement of 5-aryl-3-[2-(piperidin-1-yl)ethyl]-1,2,4-oxadiazoles upon exposure to water and HCl. Chem. Heterocycl.Compd. 2018, 54, 643–649. DOI: 10.1007/s10593-018-2321-z
  10. Li J.J.BoultonKatritzky rearrangement. In: Name Reactions. Springer, Cham, 2014. https://doi.org/10.1007/978-3-319-03979-4_34
  11. Korbonits, D.; Kanzel-Szvoboda, I.; Horváth, K. Ring transformation of 3-(2-aminoaryl)-1,2,4-oxadiazoles into 3-acylaminoindazoles; extension of the Boulton–Katritzky scheme. Chem. Soc., Perkin Trans.19821, 759-766. DOI: 10.1039/P19820000759 
  12. Kayukova, L.A.; Praliyev, K.D.; Myrzabek, A.B.; Kainarbayeva, Zh.N. Arylsulfochlorination of β-aminopropioamidoximes giving 2-aminospiropyrazolylammonium arylsulfonates. Chem. Bul.2020,69, 496–503. DOI: 10.1007/s11172-020-2789-4
  13. Kayukova, L.A.; Praliev, K.D.; Akhelova, A.L.; Kemel’bekov, U.S.; Pichkhadze, G.M.; Mukhamedzhanova, G.S.; Kadyrova,D.M.; Nasyrova,S. Local anesthetic activity of new amidoxime derivatives. Chem. J. 2011, 45(8), 468‒471. DOI: 10.1007/s11094-011-0657-0
  14. Kayukova, L.A.; Jussipbekov, U.; Praliyev, K. Amidoxime Derivatives with Local Anesthetic,     Antitubercular, and Antidiabetic Activity. IntechOpen: Open Access books; In book: Heterocycles    - Synthesis and Biological Activities; Published December, 19, 2019.
  15. Kayukova, L.A.; Uzakova, A.B.; Baitursynova,G.P.; Dyusembaeva, G.T.; Shul’gau, T.; Gulyaev, A.E.;  Sergazy, Sh.D. Inhibition of α-Amylase and α-Glucosidase by Newβ-Aminopropionamidoxime Derivatives.Pharm. Chem. J. 2019, 53, 129–133. DOI: 10.1007/s11094-019-01966-5
  16. Russia No. 2684779; Application No. 2017102282. Priority of invention 02/05/2016. The use of a derivative of beta-morpholinopropioamidoxime as an antidiabetic agent / Kayukova, L.A.;Praliev, K.D.; Dyusembaeva, G.T.; Gulyaev, A.E.; Shulgau, Z.T.; Sergazy, Sh.D.; Nurgozhin, T.S. Date of state registration in the State Register of Inventions of the Russian Federation on April 15, 2019.

(4) See all schemes. Authors should not use roman numerals to list these schemes.

Answer: Taking into account the wishes of the reviewer, the authors, firstly, revised the number of schemes, reducing them by 3, and also took into account the need for roman numbering of schemes.

(5) See page 5, lines 152-153. Authors mention that “This indicates the hydrolysis of 1,2,4-oxadiazoles 4а‒е under the influence of air moisture”. It should be improved to clarify the formation of compounds 5а‒е from 4а‒е. For instance: This indicates that the hydrolysis of 1,2,4-oxadiazoles 4а‒е under the influence of air moisture affording О-aroyl-(β-morpholin-1-yl)propioamidoximes, followed by a Boulton–Katritzky rearrangement to lead unexpected 2-amino-8-oxa-1,5-diazaspiro[4.5]dec-1-en-5-ium salts.

Answer: Thanks to the reviewer for pointing out the need to highlight an important point when considering the material of the article.  But the authors adhere to a different view of the mechanism of the Boulton-Katritsky rearrangement of 1,2,4-oxadiazoles, which we described earlier in the case of the rearrangement of 5-aryl-3-[2-(piperidin-1-yl) ethyl]-1,2,4-oxadiazoles with the formation of spiropyrazolinium benzoates and chloride. It can be represented as a series of protonation, proton transfer, and nucleophilic attack steps, effectively constituting hydrolysis during the reaction of 1,2,4-oxadiazoles 4a–e with water and wet HCl [9].

5а-е

As we have proved in this article, 1,2,4-oxadiazoles 4a–e are recorded spectroscopically (FT-IR and NMR spectral data) and they are the initial ones during hydrolysis to pyrazolinium compounds.

Taking into account this our representation, the Boulton-Katritsky rearrangement mechanism 4ae5ae, and 4a‒e6 can be represented as a sequence of protonation, proton transfer and nucleophilic attack steps, representing hydrolysis during the reaction of 1,2,4-oxadiazoles 4ae with water and wet HCl [9].

This point of view of the authors is expressed on the lines 128-130 of the revised article: “This indicates the hydrolysis of 1,2,4-oxadiazoles 4ае by the way  of Boulton-Katritsky rearrangement to spiropyrazolinium compounds 5ае under the influence of air moisture”;

and on the lines 191-196 of the revised article: “As we have proved in this article, 1,2,4-oxadiazoles 4ae are recorded by physical and chemical data and spectroscopically (FT-IR and NMR spectral data) and they are the initial ones during hydrolysis to pyrazolinium compounds. The Boulton-Katritsky rearrangement mechanism 4ae5ae, and 4ae6 can be represented as a sequence of protonation, proton transfer and nucleophilic attack steps, representing hydrolysis during the reaction of 1,2,4-oxadiazoles 4a-e with water and wet HCl in the same way as we indicated for piperidine derivatives [9]”.

 (6) See Scheme 8. The reaction for the formation of compounds 4ае from 3ае is clearly reversible. The arrow should be changed and also added only one molecule of water.     

Answer: The transition of 1,2,4-oxadiazoles 4а-е to compounds 3а-е is potentially possible, but we did not observe it. The presence of moisture in solutions 4a-e leads not to compounds 3a-e, but to compounds 5a-e. This, obviously, indicates a greater thermodynamic advantage of compounds 5а-е in comparison with compounds 3а-е. 1,2,4-Oxadiazoles 4а-е are probably metastable formations and in the presence of moisture undergo a transition to 5а-е.

 Scheme VIII. Synthetic path to 5-substituted phenyl-3-[2-(morpholin-1-yl)ethyl]-1,2,4-oxadiazoles (4a‒e), and2-amino-8-oxa-1,5-diazaspiro[4.5]dec-1-en-5-ium benzoates and сhloride hydrates(5a‒e, 6).

Counting of the oxygen atoms in the transition from compounds 4ae to compounds 5ae indicates that the latter structures differ from the previous ones by 2 oxygen atoms, which are included in the required 2 water molecules during hydrolysis.

Thus, the authors see no reason to indicate the equilibrium between 3a-e and 4a-e and to remove one water molecule in Scheme VIII or Scheme V in the revised text of the article.

 (7) See lines 156-168. It is important to unify these paragraphs. The explanation is disorganized.   

Answer: In accordance with the request of the reviewer, the specified text has been changed. (lines 131-143 of the revised article).

Further in this work, we investigated the conditions for the Boulton-Katritsky rearrangement of 5-substituted phenyl-3-[2-(morpholin-1-yl)ethyl]-1,2,4-oxadiazoles (4ae) under the deliberate establishment of hydrolysis conditions: (i) DMF with the two equivalent amount of water when heated to 60‒70 °C; (ii) alcohol / ethereal HCl mixture in the presence of air moisture (Scheme V). As can be seen from the Table 1, the heating time has an increased value (40 h) for electron-donor substituents in the phenyl ring of 1,2,4-oxadiazoles 4а, b in comparison with 1,2,4-oxadiazoles with an unsubstituted phenyl ring and with a phenyl ring having electron-withdrawing substituents – 4c‒e (25 h). Spiropyrazolinium compounds е were obtained after evaporation of DMF in an oil pump vacuum, treatment of the residue with acetone with the isolation of rearranged products 5а‒е and their recrystallization from 2-PrOH. In the case of the action of ethereal HCl solution on alcohol solutions of 1,2,4-oxadiazoles 4ae in all cases,  2-amino-8-oxa-1,5-diazaspiro[4.5]dec-1-en-5-ium chloride (6) and the corresponding benzoic acids were isolated.

(8) See table 1. (lines 145-148 of the revised article). The substituent (X) should be included. It could be re-structured. For instance, Rf and Mw values are not relevant.

Table 1. Physicochemical data of 5-substituted phenyl-3-[β-(morpholin-1-yl)ethyl]-1,2,4-oxadiazoles (4ae) and spiropyrazolinium compounds (5ae) fixed in the conditions for obtaining of single crystals for XRD analysis and in condition (i).

Compd

Yield, %

m.p., °С

Rf *

MW

Compd

t, h**

Yield, %

m.p., °С

Rf*

 MW

4a

70

230

0,71

289,33

5a

40

71

248

0,80

325,33

4b

70

220

0,62

273,33

5b

40

63

235

0,75

309,33

4c

95

216

0,66

259,31

5c

25

53

220

0,75

295,31

4d

75

224

0,67

338,20

5d

25

86

240

0,77

374,20

4e

90

190

0,62

293,75

5e

25

48

200

0,70

329,75

* Rf determined in the system ethanol:benzene, 3:1; **the heating time of 1,2,4-oxadiazoles 4ae in DMF+2H2O.

Answer: The authors agree with the reviewer that the column with the substituent X should be introduced into Table 1. At the same time, we believe that the column with Rf should not be excluded, since in the discussion of the results these values are indicative for the spiropyrazolinium series ‒ 5ae; they are larger than the Rf values of 1,2,4-oxadiazoles 4ae. Thus, the final view of Table 1 will be as follows:

Compd

X

Yield, %

m.p., °С

Rf *

Compd

t, h**

Yield, %

m.p., °С

Rf*

4a

p-CH3O

70

230

0,71

5a

40

71

248

0,80

4b

p-CH3

70

220

0,62

5b

40

63

235

0,75

4c

H

95

216

0,66

5c

25

53

220

0,75

4d

p-Br

75

224

0,67

5d

25

86

240

0,77

4e

m-Cl

90

190

0,62

5e

25

48

200

0,70

* Rf determined in the system ethanol:benzene, 3:1; **the heating time of 1,2,4-oxadiazoles 4ae in DMF+2H2O.

(9) See page 6, lines 178-186. These analysis should be improved. However, the information of the lines 195-208 is very interesting. It should be unified.  

Answer: 178-186 and 195-208. (171-175 and 176-189 lines of the revised article). In accordance with the wish of the reviewer, the authors give an extended discussion of the NMR spectra (1H and 13C) of compounds 4a ‒ e and 5a ‒ e.

It should be noted that the 1H NMR spectrum of 1,2,4-oxadiazole 4a‒e were recorded immediately with isolation; if the 1H NMR spectra were recorded after 1‒2 weeks, then the emergence and increase in the intensity of the NH2 group signal of the rearranged spiropyrazolinium products 5ae in the region of δ 7.51-7.7.57 ppm was observed. It indicates a transition of 1,2,4 -oxadiazole to the spiropyrazolinium compounds 4a‒e®5ae in the presence of air moisture.

Comparison of the NMR spectra (1H and 13C) of compounds 4ae and 5ae shows almost no differences in the regions characteristic of the groups of protons and carbon atoms of the structures of 5-substituted phenyl-3-[2-(morpholin-1-yl)ethyl]-1,2,4-oxadiazoles (4ae) and of 2-amino-8-oxa-1,5-diazaspiro[4.5]dec-1-en-5-ium benzoates (5ae). So, protons of α-СH2 and β-СH2 groups in the first case give signals in the range d 3.15‒3.17 ppm and  d 3.65‒3.66 ppm and in the region d 3.15‒3.19 ppm and d 3.64‒3.66 ppm  ‒ in the second. The signals of the aromatic protons of para-substituted 1,2,4-oxadiazoles 4a, b, d and spiropyrazolinium benzoates 5a, b, d have the form of two symmetric doublets with the spin-spin coupling constant J equal 7.5 and 8.0 Hz. Aromatic protons of 1,2,4-oxadiazoles and spiropyrazolinium benzoates with unsubstituted and meta-substituted phenyl rings have multiplet signals at d 7.23‒7.83 ppm (4c) and 7.22‒7.80 ppm (4e) and 7.25‒7.78 ppm. (5c, e). Proton-containing substituents CH3O and CH3 have singlet signals with an intensity of 3 protons at  d 3.73 and 2.27 ppm for compounds 4a, b and 5a, b.

A distinctive feature of 1H NMR spectra of benzoates 5ae from the spectra of 1,2,4-oxadiazoles 4ae is the presence of NH2 proton signal with the integral intensity of 2H at d 7.51–7.57 ppm.

 1H NMR spectrum of spirocompound 6 contained the triplet proton signals of α- and β-CH2 groups at 3.16 and 3.68 ppm and signals of N(+)(CH2)2(CH2)2О and N(+)(CH2)2(CH2)2О groups at d 3.40 and 3.92 ppm, respectively; no aromatic proton signals were observed.

Of the remarkable features of the 1H NMR spectra of compounds 4a‒e, 5a‒e, and 6, is that the axial and equatorial protons of the methylene groups located at the nitrogen atom of the morpholine ring give independent multiplet signals. In one case these signals are superimposed with the common signal of two groups methylene protons located at the oxygen atom of the morpholine ring at δ 3.91‒3.93 ppm with a total intensity of six protons and in the other case have a multiplet signal at ~ δ 3.40 ppm intensity of two protons. The diastereotopicity of discussed geminal protons of compounds 4a‒e, 5a‒e, and 6 is associated with a dynamic cause due to slow rotation of the morpholine heterocycle. The effect of hindered inversion of six-membered heterocycles, with a chair-like conformer with fixed positions of the axial and equatorial protons being predominant, in the 1H NMR spectra is a known fact reported in reference data [21]. Besides the diastereotopicity of these geminal protons of compounds 5a‒e, and 6 is associated with asymmetry due to the presence of the spirocyclic system. 

In the 13C spectra 4ae and 5ae, all signals of aliphatic and aromatic protons were recorded in the expected regions.

(10) See figures 1 and 2. It should be included in the supplementary material. A properly explanation and organization of NMR information may avoid the inclusion of figures 1 and 2 in the manuscript. However, If you want to include NMR spectra, it should be improved (i.e. resolution and size).

Answer: In response to the comment of the reviewer (10), the authors came to the conclusion that instead of figures in the text of the article it is better to use a proper explanation of NMR information and thereby to avoid the inclusion of the figures 1 and 2 in the manuscript. Accordingly, the text above in the response to Remark 9 should replace Figures 1 and 2 and their explanations.

 (11) See table 2. The MIC values in μg/ml should be included instead of symbols like +, ++ and +++. It is important to check articles related to medicinal chemistry (i.e., J. Med. Chem., Eur. J. Med. Chem., Bioorg. Chem., among others) in order to see the presentation of these information. However, the explan

Answer: The authors agree with the reviewer and provide the revised table 2 and its discussion (lines 198-214 of revised paper).

2.2 In vitro antitubercular screening of 2-amino-8-oxa-1,5-diazaspiro[4.5]dec-1-en-5-iumbenzoates and сhloride (5a‒e, 6).

In vitro antitubercular bacteriostatic activity of spiropyrazolinium compounds 5ae and 6 on drug-sensitive (DS) and multidrug-resistant (MDR) of M. tuberculosis (MTB) strains was studied using the method of serial dilution on the liquid Shkolnikova medium (Table 2).

Table 2. In vitro antitubercular activity (MIC) of 2-amino-8-oxa-1,5-diazaspiro[4.5]dec-1-en-5-ium benzoates and chloride (5ae, 6), μg/ml* on DS (H37Rv ) and MDR (I) strains of MTB.

Compd

5a

5b

5c

5d

5e

6

Rifampicin

MIC,  μg/ml

H37Rv

50

10

20

100

100

1

1

I

50

50

50

100

100

2

2

* In the upper line of MIC values the activity on the DS of MTB strains (H37Rv) is shown; in the lower ‒ on the wild MDR strains (I), isolated from the patient, resistant to rifampicin and isoniazid.

A number of compounds 5ae on DS and MDR MTB strains exhibits an average antitubercular activity of 10–100 μg/ml. Moreover, an improvement in activity is observed with a decrease in MIC to 10 μg/ml (5b); 20 μg/ml (5c) and 50 μg/ml (5a) on the DS MTB strains and up to 50 μg/ml on the MDR MTB strains for the compounds 5ac containing donor substituents in the phenyl ring or with an unsubstituted phenyl ring. Spiropyrazolylammonium chloride 6 demonstrates high in vitro  antitubercular activity equal to activity of the basic antitubercular drug of the first row rifampicin: on the DS strain as low as 1 μg/ml; on the wild MDR strain ‒ 2 μg/ml.

 (12) See material and methods. It is important to see a paper recently published in Molecules related to organic synthesis. For instance: (a) The 13C NMR data should be reported with only one instead of two decimals after the point, (b) the deuterated solvent and frequency should be included [i.e. 1H NMR (400 MHz, CDCl3): d =] and/or [i.e. 13C{1H} NMR (101 MHz, CDCl3): d =], (c) the solvent or mixture of solvents for recrystallization process should be included for the corresponding compound, (d) the symbol of the coupling constant (J) should be in italic, and generally (e) the 1H NMR data should be reported like xx.xx (multiplicity, coupling constant, integration, assignation).    

Answer: In accordance with the wishes of the reviewer, the authors made changes in the Material and methods section. But the presentation of 13C NMR spectra in papers recently published in Molecules related to organic synthesis: Molecules 2020, 25, 5924; doi: 10.3390 / molecules25245924; Molecules 2020, 25, 5779; doi: 10.3390 / molecules25245779 allows the use of two decimal places.

(13) See material and methods. It is not necessary to repeat the procedure of each compound. A general procedure should be included for each family of synthesized compounds. It is important to see a paper recently published in Molecules related to organic synthesis.

Answer: The authors took into account the reviewer's feedback and added general procedures for the synthesis of compounds 4, 5, 6 to the section on materials and methods (lines 341-461 of the revised article).

(14) See material and methods. See Table 4. It should be included in the supplementary material.

Answer: As requested by the reviewer, the authors have included Table 4 in the supplementary material.

 (15) I did not find the supplementary material. All NMR spectra of synthesized compounds 4, 5 and 6 should be added and properly processed including the integration of each signal (1H NMR) and chemical shifts (both 1H and 13C{1H} NMR) after previous calibration with TMS or deuterated solvent.  

Answer: In accordance with the wish of the reviewer, the authors performed the required representation of the NMR spectra of compounds 4, 5 and 6 and included them in the supplementary material.

(16) I did not find the conclusion section. It should be included according to the guideline of this journal.    

Answer: At the request of the reviewer, the authors included a conclusion section in the article (lines 492-508 of the revised article).

Date of answer: 16 Jan 2021                                                                                                          

From the authors: L.A. Kayukova

Round 2

Reviewer 3 Report

I agree that manuscript is suitable to publish in Molecules.